# Activation of Metabotropic Glutamate Receptor (mGlu_2_) and Muscarinic Receptors (M_1_, M_4_, and M_5_), Alone or in Combination, and Its Impact on the Acquisition and Retention of Learning in the Morris Water Maze, NMDA Expression and cGMP Synthesis

**DOI:** 10.3390/biom13071064

**Published:** 2023-06-30

**Authors:** Joanna M. Wierońska, Paulina Cieślik, Grzegorz Burnat, Leszek Kalinowski

**Affiliations:** 1Maj Institute of Pharmacology Polish Academy of Sciences, Smętna 12, 31-343 Kraków, Poland; pa.cieslik@gmail.com (P.C.); burnat@if-pan.krakow.pl (G.B.); 2Department of Medical Laboratory Diagnostics—Fahrenheit Biobank BBMRI.pl, Medical University of Gdansk, 7 Debinki Street, 80-211 Gdansk, Poland; leszek.kalinowski@gumed.edu.pl; 3BioTechMed Centre, Department of Mechanics of Materials and Structures, Gdansk University of Technology, 11/12 Gabriela Narutowicza Street, 80-233 Gdansk, Poland

**Keywords:** Morris water maze, schizophrenia, spatial learning, muscarinic receptors, mGlu receptors, cGMP, NMDA

## Abstract

The Morris water maze (MWM) is regarded as one of the most popular tests for detecting spatial memory in rodents. Long-term potentiation and cGMP synthesis seem to be among the crucial factors involved in this type of learning. Muscarinic (M_1_, M_4_, and M_5_ receptors) and metabotropic glutamate (mGlu) receptors are important targets in the search for antipsychotic drugs with the potency to treat cognitive disabilities associated with the disorder. Here, we show that muscarinic receptor activators (VU0357017, VU0152100, and VU0238429) and an mGlu_2_ receptor activator, LY487379, dose-dependently prevented the development of cognitive disorders as a result of MK-801 administration in the MWM. The dose-ranges of the compounds were as follows: VU0357017, 0.25, 0.5, and 1 mg/kg; VU0152100, 0.05, 0.25, and 1 mg/kg; VU0238429, 1, 5, and 20 mg/kg; and LY487379, 0.5, 3, and 5 mg/kg. The co-administration of LY487379 with each of the individual muscarinic receptor ligands showed no synergistic effect, which contradicts the results obtained earlier in the novel object recognition (NOR) test. MWM learning resulted in increased cGMP synthesis, both in the cortex and hippocampi, when compared to that in intact animals, which was prevented by MK-801 administration. The investigated compounds at the highest doses reversed this MK-801-induced effect. Neither the procedure nor the treatment resulted in changes in GluN2B-NMDA expression.

## 1. Introduction

Cognitive impairments accompany many psychiatric diseases and impact everyday functioning [1]. At present, no satisfactory drugs are available to ameliorate or endure cognitive decline in humans [1]. Novel solutions are highly needed.

Glutamatergic and cholinergic receptors are considered important regulators of learning and memory processes. In particular, their important role in the regulation of the glutamate–NO–cGMP axis and long-term potentiation (LTP) was described, with the process being critical in cognition [2,3,4,5]. In recent years, high hopes have been pinned on activation of the mGlu_2_ subtype or cholinergic receptors as potent antipsychotic drugs [5,6,7,8]. The clinical trials with mGlu_2_ potentiators end up with mixed results [9,10,11,12]. Reports indicate that the ligands may be more effective when administered under certain conditions to a selected subpopulation of patients [13,14]. Their efficacy can be also improved by add-on therapy, as shown in variety of our recent papers [15,16]. Muscarinic receptor activators are one of the candidates.

Among the five muscarinic receptor subtypes, M_1_, M_4_, and M_5_ are highly expressed in the central nervous system (CNS), in contrast to M_2_ and M_3_ receptors with significant expression also in peripheral tissues, where their stimulation can evoke off-target effects [17,18,19]. These effects controversially arose in clinical trials with xanomeline, a non-selective muscarinic receptor agonist, which was investigated as a memory-improving compound in schizophrenic patients [20,21,22] and has been shown to improve positive, negative, and cognitive symptoms after just one week of treatment.

Selective activators of M_1_, M_4_, and M_5_ receptors were recently shown to be effective in animal models of schizophrenia. M_4_ receptor activators were effective in the models of all symptoms of schizophrenia, while M_1_ and M_5_ activators were predominantly effective in MK-801-induced amnesia in the novel object recognition test [15,16,23]. Thus, it could be hypothesized that muscarinic and mGlu_2_ receptor ligands could exclusively function in animal models of learning and memory. To further investigate this hypothesis, the Morris water maze was used. This is an assay for hippocampus-dependent spatial learning and memory, where pharmacological manipulations disrupt activity-dependent plasticity and prevent spatial learning [24,25,26]. High reliability across a wide range of tank configurations and testing procedures and extensive evidence of its validity as a measure of reference memory are important characteristics of the test [26]. The activity of muscarinic and mGlu_2_ receptor activators in MK-801-treated mice has not been studied in this paradigm so far. Thus, continuing our previous research, the activity of VU0357017 (M_1_ allosteric agonist), VU0152100 (M_4_ positive allosteric modulator), VU0238429 (M_5_ positive allosteric modulator), and LY487379 (mGlu_2_ positive allosteric modulator) in the MWM, and subsequently, the combination of mGlu_2_ PAM and the particular muscarinic receptor ligands, were investigated. Concomitantly, the expression of NMDA receptors and the level of cGMP, two factors crucial for LTP formation [2,3], were investigated in the frontal cortices and hippocampi of mice.

## 2. Materials and Methods

### 2.1. Animals

Male CD-1 mice were used in the study (Charles River). The animals weighted ~20 g at the time of arrival. After an acclimatization period and several days of handling and experimental procedures, their body weight increased up to 30 g (this weight corresponds to approximately 4–5 weeks of age). The animals were kept under standard conditions (12:12 light:dark cycle, 22 ± 1 °C, 55 ± 10% humidity) with water and food available ad libitum. All drugs were administered intraperitoneally (i.p.) at a volume of 10 mL/kg. The procedures were conducted in accordance with the European Communities Council Directive of 22 September 2010 (2010/63/EU) and Polish legislative acts concerning animal experimentation and were approved by the II Local Ethics Committee by the Maj Institute of Pharmacology, Polish Academy of Sciences in Krakow.

### 2.2. Drug Preparation and Administration

MK-801 and VU0357017 were dissolved in 0.9% saline. VU0152100 and VU0238429 were sonicated in small amounts of 10% Tween 80 in 0.9% NaCl and titrated to appropriate volumes. LY487379 was dissolved in a small amount of DMSO (final concentration: <0.1%) and then adjusted to a proper volume with 0.9% saline. When the administration of the tested compounds was omitted (control and MK-801 groups), the animals received appropriate vehicles. The doses used in the behavioral experiments were based on our previous studies [15,16,23].

In the first set of experiments, drugs were administered alone to determine the dose-dependent effects on MK-801-induced dysfunctions in the Morris water maze test.

The M_1_ allosteric agonist, VU0357017, was administered at doses of 0.25, 0.5, and 1 mg/kg. The PAM of the M_4_ receptor, VU0152100, was administered at doses of 0.05, 0.25, and 1 mg/kg. The PAM of the M_5_ receptor, VU0238429, was administered at doses of 1, 5, and 20 mg/kg, and the positive allosteric modulator of mGlu_2_ receptors, LY487379, was administered at doses of 0.5, 3, and 5 mg/kg.

In the second part of the experiment, the activity of the simultaneous administration of LY487379 with each of the muscarinic activators was tested. The following combinations were administered: LY487379 + VU0357017, LY487379 + VU0152100, and LY487379 + VU0238429. Each pair of compounds was administered at the following doses: inactive doses of both compounds, intermediate doses of both compounds, highest doses of both compounds (Table 1).

### 2.3. Morris Water Maze (MWM)

The MWM test was performed according to Sałat et al. [27] with minor modifications. Mice were trained to find a hidden platform (10 cm in diameter, ca. 1 cm beneath the water surface) in a circular pool (120 cm in diameter) using spatial cues. Visual cues were consistent during the experiments and included geometric figures placed in the proximity of the maze, with the experimenters and differently colored walls surrounding the maze. The water temperature was maintained at 22–23 °C. The pool was divided into 4 quadrants, namely, NE, NW, SE, and SW, and the platform was located in the NE quadrant (Figure 1). ANY-maze (Stoelting) was used to track the animals during the MWM.

The acquisition phase consisted of 5 training days with 4 training trials per day. During each trial, the animal was gently placed in the pool and was allowed to explore it for up to 60 s in order to find the hidden platform. Starting locations (1–4, Figure 1) were selected in a pseudorandom manner for each training day. The drugs, either alone or in combination, were administered 30 min before MK-801, which was administered 30 min before the first training trial each day. During this phase, escape latency (in seconds), defined as the time required for animals to reach the hidden platform, was measured (means ± SEMs are shown for each group on each training day).

The retention trial was performed on the day after the acquisition phase and consisted of one 60 s trial. During the trial, the platform was removed from the pool. No drugs were administered in this phase. During the retention trial, several parameters were measured, grouped as NE zone-related parameters (number of entries into the zone, time in the zone, distance travelled in the zone, distance travelled until 1st entry into the zone, latency to 1st entry into the zone, and the path efficiency to 1st entry into the zone) and platform zone (PZ)-related parameters (number of entries into the PZ, time in the PZ, distance travelled until 1st entry into the PZ, latency to 1st entry into the PZ, and the path efficiency to 1st entry into the PZ).

### 2.4. Synaptosome Preparation and Western Blotting

After the MWM, mice receiving the vehicle, MK-801, and the highest doses of muscarinic and mGlu_2_ receptor ligands were sacrificed, and the frontal cortices and the whole hippocampi were collected. The tissues were stored at –80 °C until analysis.

The fractions were prepared according to the procedure described by Pochwat et al. [28] with minor modifications. Samples were homogenized in cold lysis buffer (0.32 M sucrose, 20 mM HEPES (pH 7.4)), with 1 mM EDTA, 5 mM NaF, 1 mM NaVO_3_, and a protease inhibitor cocktail (Thermo Fisher Scientific, Waltham, MA, USA). Homogenates were then centrifuged at 2800× *g* rpm for 10 min at 4 °C. The obtained supernatants were then centrifuged at 12,000× *g* rpm for 10 min at 4 °C, and the pellets (crude synaptosomal fractions) were sonicated in protein lysis buffer containing 50 mM Tris HCl (pH 7.5), 150 mM NaCl, 1% Triton X-100, 0.1% SDS, 2 mM EDTA, 1 mM NaVO_3_, 5 mM NaF, and protease inhibitor cocktail. Protein concentrations were measured using a BCA kit (Thermo Fisher Scientific).

A total of 30 µg of total protein from each sample was mixed with Laemmli buffer (Bio-Rad, Hercules, CA, USA) containing 2.5% β-mercaptoethanol (Bio-Rad) to a final volume of 12 or 16 µL. Samples were separated on 8% SDS-PAGE gels, transferred to nitrocellulose membranes, and blocked overnight at 4 °C (1% blocking solution; BM Chemiluminescence Western Blotting Kit (Mouse/Rabbit)). Membranes were then incubated with the following primary antibodies: recombinant rabbit monoclonal antibody anti-NMDAR2B (1:1500, 2 h, RT) (Abcam, Cambridge, UK) and mouse monoclonal antibody anti-β-actin (1:7500, 1 h, RT) (Sigma Aldrich, St. Louis, MI, USA). The membranes were washed in TBS-T (6 × 5 min) and incubated for 1 h at RT with anti-mouse IgG-peroxidase conjugated/anti-rabbit IgG-peroxidase conjugated antibody (1:12,500; BM Chemiluminescence Western Blotting Kit (Mouse/Rabbit)), washed again, and developed (BM Chemiluminescence Western Blotting Kit (Mouse/Rabbit)). Signals were detected using a GeneGnome XRQ analysis system (SYNGENE, Hyderabad, India) and analyzed using Image Gauge V4.0.

### 2.5. cGMP ELISA

The samples were homogenized in 5% TCA in water and centrifuged at 3000× *g* for 5 min. The levels of acetylated cGMP in the supernatant were quantified using an ELISA assay (Cayman Chemicals, Ann Arbor, MI, USA) according to the manufacturer’s instructions. Under experimental conditions, the standard curve range was 0.023–3 pmol/1000 μL.

### 2.6. Statistics

Statistical significance was determined using a Student’s *t*-test (control vs. MK-801), one-way ANOVA followed by Tukey’s post hoc comparison (cGMP—“control” groups), or Dunnet’s post hoc comparison (MWM, cGMP—treatment). If the assumption of normal distribution was not met, the data were analyzed using U Mann–Whitney tests or Kruskal–Wallis tests. Regarding the acquisition phase, only data from the last day were analyzed using one-way ANOVA or a Kruskal–Wallis test to assess if at the end of the training procedure there were significant differences between MK-801 and treatment groups. The differences between control and MK-801 groups were analyzed using a Student’s *t*-test or U Mann–Whitney test.

The data were analyzed using TIBCO Statistica (v.13.3) or GraphPad Prism (v.9.4.1) and presented as the mean ± SEM.

## 3. Results

### 3.1. Morris Water Maze

#### 3.1.1. Acquisition Phase

The administration of MK-801 impaired the ability of mice to learn the location of the hidden platform in all performed experiments, which was measured as decreased escape latency when compared to controls (e.g., Z = 3.212, *p* < 0.01).

The M_1_ allosteric agonist, VU0357017, prevented MK-801-induced effects at the dose of 0.25 mg/kg (H(3) = 7.848, *p* < 0.05). The doses of 0.5 and 1 mg/kg were ineffective. The PAM of M_4_ receptor, VU0152100, prevented MK-801-induced effects at the dose of 1 mg/kg (H(3) = 7.954, *p* < 0.05), and activity at the doses of 0.05 and 0.25 mg/kg was not statistically significant. The PAM of the M_5_ receptor, VU0238429, was active at the doses of 1 and 20 mg/kg (H(3) = 13.503, *p* < 0.01), but not at 5 mg/kg. The positive allosteric modulator of mGlu_2_ receptors, LY487379, reversed MK-801-induced impairments at the doses of 3 or 5 mg/kg (H(3) = 11.136, *p* < 0.05), but not at 0.5 mg/kg.

The concomitant administration of LY487379 and VU0357017 reversed MK-801-induced disruptions with the combinations of the low (0.5 + 0.25 mg/kg) and highest (5 + 1 mg/kg) (H(3) = 11.8545, *p* < 0.01) doses. The effectiveness of the combination of LY487379 and VU0152100 was statistically significant for the intermediate doses (3 + 0.25 mg/kg) (H(3) = 8.4305, *p* < 0.05). No effect was observed when LY487379 was administered with VU0238429 (H(3) = 4.7644, *p* = 0.1901) in any combination.

Graphs related to the results from the acquisition phase can be found in the Appendix A.

#### 3.1.2. Retention Trial

The administration of MK-801 impaired spatial memory in terms of several of the parameters measured (Appendix A). When administered alone, the compounds had no impact on animals’ behavior comparing to controls. The results are shown in the Appendix A.

The administration of VU0357017 prevented some of the MK-801-induced impairments at the doses of 0.5 and 1 mg/kg. The dose of 0.25 mg/kg was ineffective (Figure 2). The administration of VU0152100 prevented two NE zone-related parameters at the doses of 0.5 and 1 mg/kg and only one parameter at the dose of 0.05 mg/kg (Figure 3). VU0238429 reversed the disruptions in three measured parameters at the dose of 20 mg/kg and one parameter at the dose of 5 mg/kg. The dose of 1 mg/kg was not effective (Figure 4). LY487379 reversed the deficits in five measured parameters at the dose of 5 mg/kg and one parameter at the doses of 0.5 and 1 mg/kg (Figure 5). Working doses are indicated on graphs presented in Figure 2, Figure 3, Figure 4 and Figure 5. Detailed statistical analysis, swimming speeds, and heat maps are shown in the Appendix A.

The simultaneous activation of mGlu_2_/M_1_ receptors prevented three MK-801-induced dysfunctions (Figure 6). The concomitant administration of LY487379 and VU0152100 (mGlu_2_/M_4_ activators) reversed disruptions in five parameters (Figure 7). The combination of LY487379 and VU0238429 (mGlu_2_/M_5_ activators) reversed disruptions in five parameters when administered at low, intermediate, or the highest doses (Figure 8). Detailed statistical analyses and working doses are shown in Appendix A.

### 3.2. GluN2B Expression

No significant changes in the level of GluN2B were observed between the trained and untrained mice receiving vehicle or MK-801, both in the FC and the hippocampus. The administration of muscarinic or mGlu_2_ ligands had no effect on the hippocampal GluN2B level in trained mice when compared with that in trained MK-801-treated mice (F_(4.45)_ = 0.4529, p = 0.7697). In the FC, only the administration of VU0357017 increased the level of GluN2B in comparison with that in trained MK-801-treated mice (F_(4.44)_ = 2.823, *p* < 0.05) (Figure 9).

### 3.3. cGMP

An elevated cGMP level was observed in the FC and hippocampi of control, MWM-trained mice when compared with that in control, untrained mice (Student’s *t*-test: FC: *t* = 4.407, df = 14, *p* < 0.0006; hippocampus: *t* = 3.56, df = 14, *p* < 0.003) (Figure 10).

The administration of MK-801 induced a decrease in the cGMP level in MWM-trained animals when compared to that in MWM-trained controls (Student’s *t*-test: FC: *t* = 3.95, df = 14, *p* < 0.0016; hippocampus: *t* = 3.06, df = 14, *p* < 0.008) (Figure 10).

The administration of VU0357017, VU0152100, VU0238429, or LY487379 ameliorated the effects of MK-801 administration on cGMP accumulation in the PFC and hippocampus. One-way ANOVA analysis revealed that the compounds not only prevented MK-801-induced effects but also induced increases in cGMP when compared to levels in control animals (Figure 10). A detailed statistical analysis is presented in Appendix A.

## 4. Discussion

In the present study, the Morris water maze test was used to evaluate the potency of the activators of muscarinic receptors (M_1_, M_4_, and M_5_) and the positive allosteric modulator of the mGlu_2_ receptor to prevent MK-801-induced spatial memory dysfunction. The experiments are a continuation of our recent work that focused on the effectiveness of muscarinic receptor ligands (and in some cases mGlu) in the models of cognitive impairment associated with schizophrenia.

Escape latency, defined as the time needed by the animal to find the platform, was the basic parameter measured during five days of the acquisition phase. In the retention trial, which was performed 24 h after the last drug administration and training, the preference to navigate to the zone in which the platform was located (NE zone) and the accuracy of finding the platform zone (PZ) were investigated.

In general, all investigated compounds reversed spatial learning deficits induced by MK-801 administration, but different impacts on the parameters were observed. LY487379, the mGlu_2_ receptor activator, was most potent, and VU0357017, the M_1_ receptor activator, was most active towards PZ-related deficits, among all muscarinic ligands. Among the investigated compounds, LY487379-treated animals explored the pool most effectively. In the present studies except latency to the 1st entry to the NE zone or PZ, a less popular measures of MWM learning were also analyzed.

In the second part of the experiment, which was also a continuation of our previous studies, the effects of the combined administration of LY487379, a PAM of the mGlu_2_ receptor, and each of the muscarinic receptor activators were measured. No synergistic effects of the combinations were observed, regardless of which pair of doses was administered. The compounds (and their combinations) failed to alter general motor activity or exploratory behavior. In the test session, no changes in swimming velocity were observed.

The results had some contradictions with previous studies in which clear synergistic effects of the combined administration of subeffective (but not the highest) doses were observed in NOR and prepulse inhibition (PPI) tests [15,16]. Pharmacokinetic studies revealed that the simultaneous administration of compounds did not induce any non-specific drug–drug interactions [15].

The observed behavioral discrepancies were most likely a result of distinct mechanisms of learning associated with particular tests of cognition. In general NOR is a test of the memory of an episode, and PPI investigates the ability to filter out the unnecessary information [29,30], in which the frontal cortex plays important roles.

The Morris water maze, similarly to other spatial learning procedures, is thought to measure hippocampal-dependent learning, and the additional difficulty of the test is that the animal must decide continuously where to go during a one-minute trial [24,25,26]. Additionally the behavior of the animals (as the reference memory) is measured after several days of training and 24 h after the last treatment. Thus, animal behavior is not directly affected by the compounds and can be considered long-term memory.

Other popular tasks for spatial learning, such as the T- or Y-maze, are more structured, forcing the animals to make only a binary decision, moving towards the right or left arm of the maze [31,32,33]. In our hands, poor activity of muscarinic ligands in the T-maze was observed [23]. The effectiveness of the highest doses was evident only after repeated (7 days) administration.

It is generally accepted that MWM deficits develop as a consequence of hippocampal dysfunction and long-term potentiation (LTP) deficits [34,35,36]. The process of LTP is essential for long-term memory storage and is regulated by a variety of mechanisms. The glutamate–NO–cGMP axis is one of the most important [3], and in this process, the activation of NMDA receptors by glutamate provokes NO synthesis and subsequent cGMP production [2]. Thus, in the present study, we additionally performed biochemical investigations aimed at establishing the impact of both the procedure and drug administration on NMDA receptor expression and cGMP synthesis.

Functional NMDA receptors are heterotetramers composed of two GluN1 and typically two GluN2 subunits. Four GluN2 isoforms (GluN2A–D) have been identified, with GluN2A and GluN2B isoforms dominating in the adult brain. GluN2A-containing receptors desensitize more rapidly than GluN2B-containing receptors, which show a much slower desensitization time course. Some evidence suggests that GluN2A preferentially distributes to the postsynaptic density, compared with GluN2B, which also distributes throughout the dendrite at extrasynaptic sites and mediates higher cognitive operations, such as working memory. Finally, a comparison of clinical phenotypes shows that GluN2A variants are commonly associated with an epileptic phenotype, but GluN2B variants are commonly found in patients with neurodevelopmental disorders, schizophrenia, and Alzheimer’s disease [37,38,39,40]. Thus, it is commonly accepted that the basic structure and functions associated with the GluN2B-NMDA receptor in psychiatric disorders can be attributed to the GluN2B subunit. Additionally, strong evidence indicates that the GluN2B-NMDA subunit plays a critical role in the induction of LTP [41]. Accordingly, the expression of GluN2B was shown in the present studies to assess the impact of the procedure and treatment on NMDA receptor expression. No changes in GluN2B-NMDA expression were observed in trained vs. non-trained animals, and no significant changes were observed between drug-treated groups, with the exception of the VU0357017-treated group, in which a slight increase in GluN2B-NMDA receptor expression was observed in the hippocampus. The biological significance of this remains to be established. In contrast, significant changes in the cGMP level were observed. Several days of training dramatically increased the cGMP level, both in the cortex and hippocampus, when compared to that in intact controls. MK-801 administration significantly prevented cGMP production induced by the training, which, at least partially, may be responsible for memory dysfunction in the MWM. The administration of all investigated compounds not only reversed these MK-801-induced deficits but also induced increases in cGMP production above the control level. The results confirm that fluctuations in cGMP production translate into learning abilities in the MWM and that the potency of the tested compounds to prevent MK-801-induced memory dysfunction is also cGMP-dependent.

Many of the mechanisms controlling the production of NO/cGMP remain unclear, but both inhibitory GABAergic systems and excitatory glutamatergic neurotransmission modulate the production of both NO and cGMP [42,43]. The majority of reports, in general, confirm that the inhibition of cGMP synthesis is linked to decreased learning in the MWM. No comparisons of cGMP levels between “normal” and trained animals have been shown to date; thus our results are the first in this field. Similarly, no data regarding the impact of muscarinic receptor activators, in particular M_1_, M_4_, and M_5_, or mGlu_2_ activators, on the cGMP level are available. Thus, our results are the first to confirm that the reversal of MK-801-induced reference memory dysfunction in the MWM is cGMP-dependent.

In summary, our results show that M_1_, M_4_, and M_5_ receptor activators prevent many aspects of MK-801-induced memory impairment in the MWM test, thus confirming the important role of these receptors in many aspects of cognition and cGMP-dependent activity. The simultaneous activation of muscarinic and mGlu_2_ receptors does not exert any synergistic effect in this test, indicating a different mechanism involved in MWM spatial learning compared to episodic memory deficits in the NOR. In contrast, short-term working memory deficits related to spatial learning are more resistant to the effects of muscarinic or mGlu_2_ receptor activators, and repeated administration of the compounds is required to observe any treatment effects. The simultaneous administration of two selected receptors at low doses is not a solution to this type of learning, as in contrast to previous studies, no add-on effects of drugs were observed.

## 5. Conclusions

Here, we show that the ligands of M_1_, M_4_, and M_5_ receptors prevented the development of MK-801-induced reference memory dysfunction in the MWM. A similar efficacy was shown for the mGlu_2_ activator LY487379. This indicates that the ligands are potent not only in preventing short-term memory dysfunction induced by MK-801 administration (as published in [15,16,23]) but also long-term reference memory, which was independent of the memory of the last training session, as investigated 24 h later. This confirms the reports of the others that investigated receptors that play important roles in LTP [5,6,7,8].

The latency of the 1st entry to the NE zone and/or PZ are most often used to determine learning abilities in the MWM. Here, we showed that some least popular measures of MWM learning can work better and/or constitute additional indicators of spatial learning.

When LY487379 was co-administered with each of muscarinic receptor ligands, no synergistic actions of the combinations were observed.

The process of learning, MK-801-induced amnesia, and its prevention mediated by the investigated ligands were cGMP-dependent, but the changes in the expression of GluN2B-NMDA subunits were not noticed, suggesting that the procedure and pharmacological manipulations putatively affected the glutamate–NO–cGMP axis and subsequent LTP, changing NMDA receptor susceptibility and functionality, but not density.

## Figures and Tables

**Figure 1 biomolecules-13-01064-f001:**
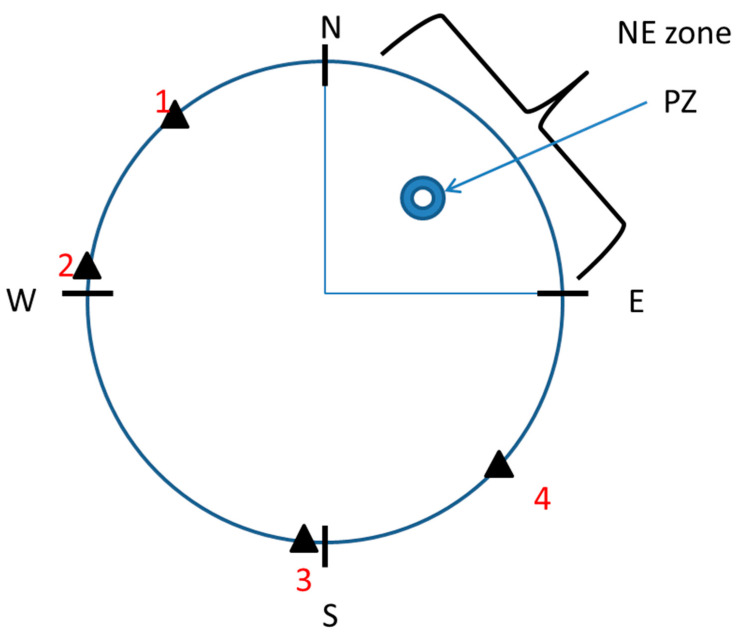
Schematic representation of the Morris water pool. PZ—platform; NE—the area inside which the platform was located; 1, 2, 3, 4—starting points.

**Figure 2 biomolecules-13-01064-f002:**
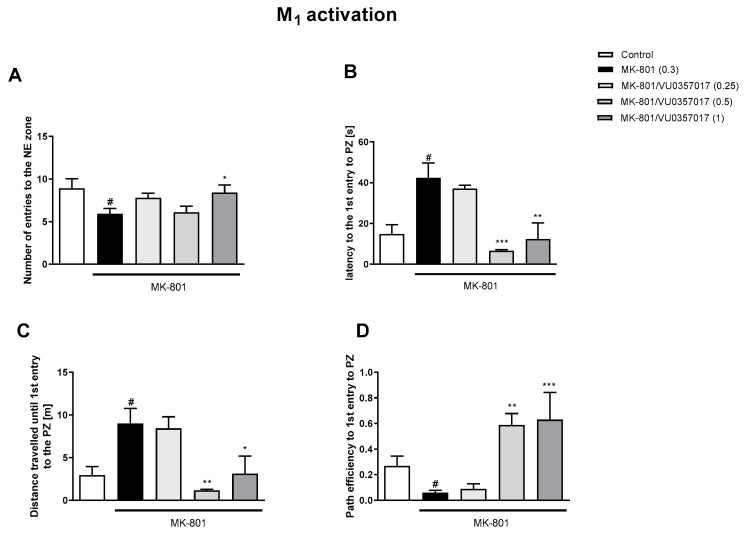
The effects of VU0357017 on MK-801-induced deficits in the retention of spatial memory in the MWM in CD1 mice. Sub-figures represent number of entries to the NE zone (**A**), Latency to the 1st entry to PZ (**B**), distance travelled until 1st entry to PZ (**C**) and pth efficiency to the 1st entry to PZ (**D**). The values are expressed as the means ± SEMs. ^#^
*p* < 0.001 vs. control group and * *p* < 0.05, ** *p* < 0.01, and *** *p* < 0.001 vs. the MK-801-treated animals (N = at least 8–10). PZ—platform zone.

**Figure 3 biomolecules-13-01064-f003:**
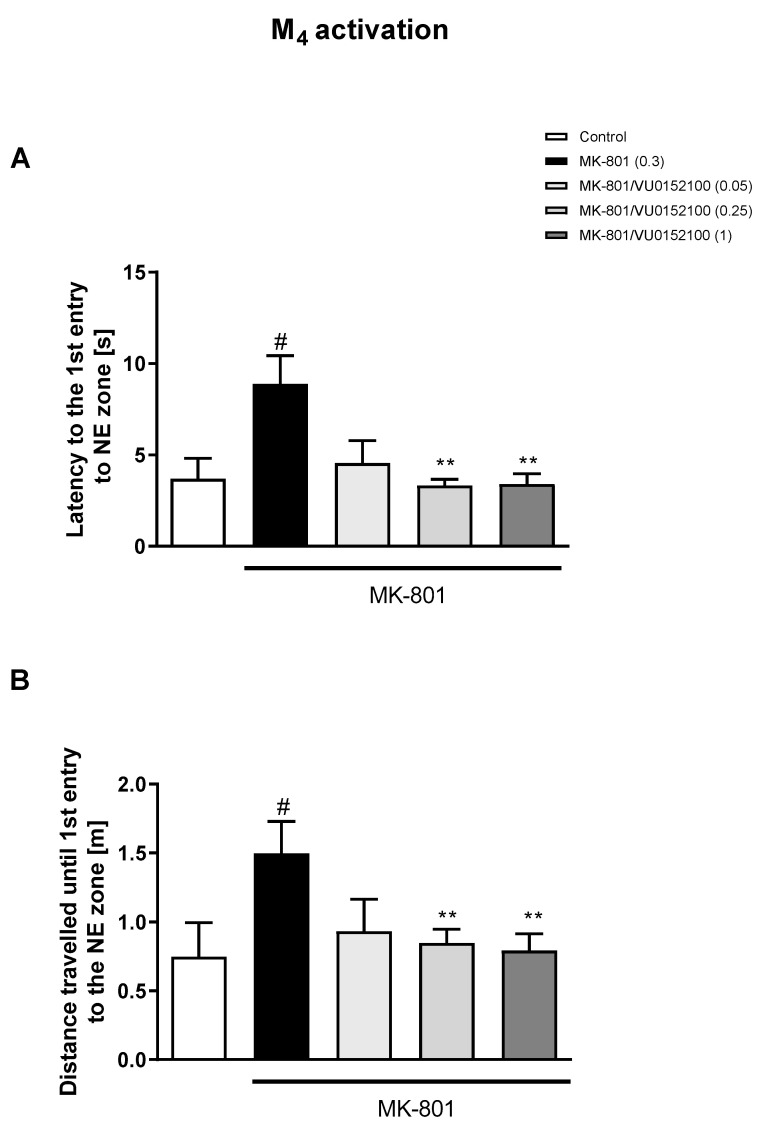
The effects of VU0152100 on MK-801-induced deficits in the retention of spatial memory in the MWM in CD1 mice. Sub-figures represent latency to the 1st entry to the NE zone (**A**) and distance traveled until the 1st entry to the NE zone (**B**). The values are expressed as the means ± SEMs. ^#^
*p* < 0.001 vs. control group and ** *p* < 0.01 vs. the MK-801-treated animals (N = at least 8–10).

**Figure 4 biomolecules-13-01064-f004:**
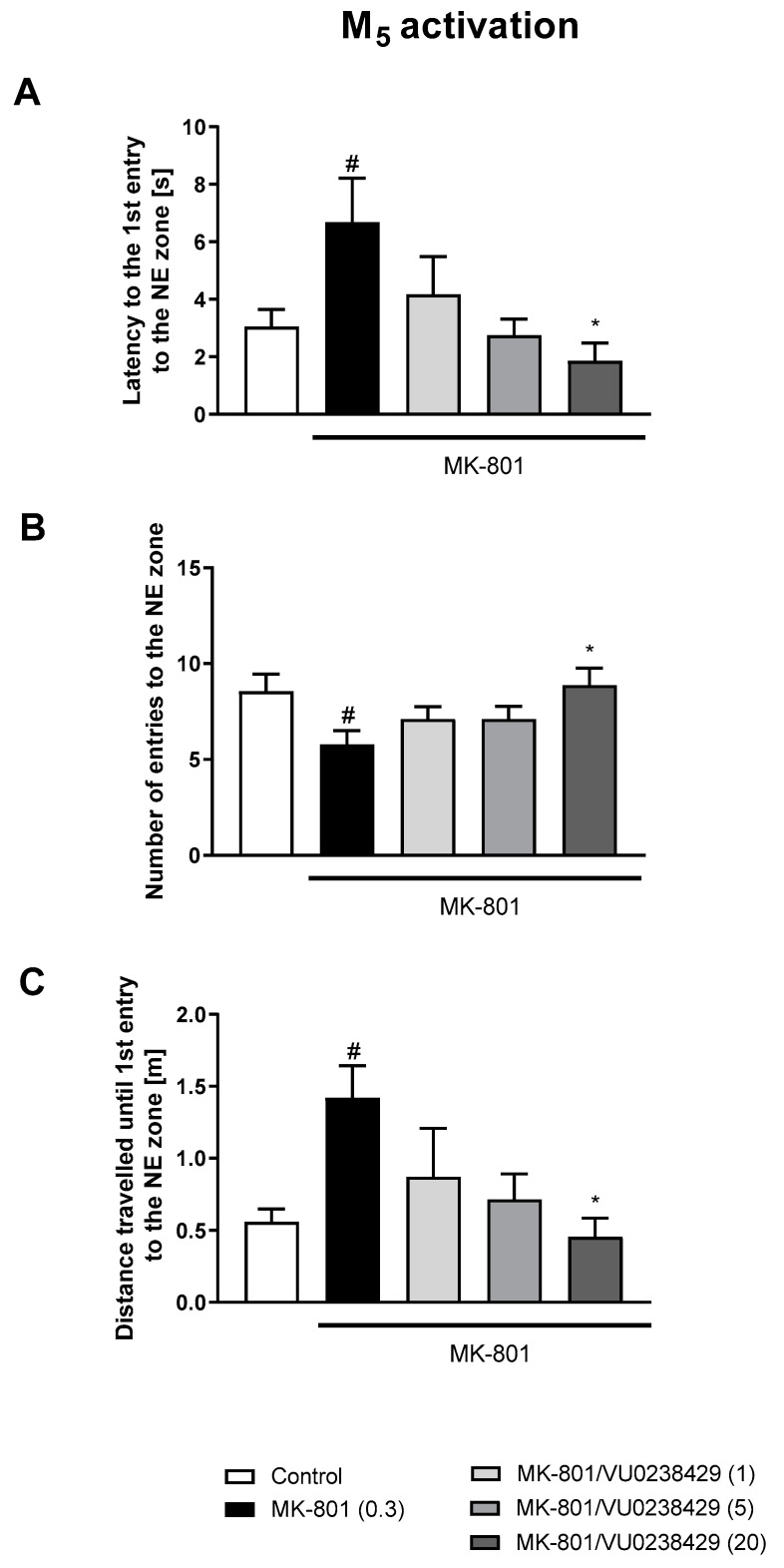
The effects of VU0238429 on MK-801-induced deficits in the retention of spatial memory in the MWM in CD1 mice. Sub-figures represent latency to the 1st entry to the NE zone (**A**), number of entries to the NE zone (**B**) and distance travelled until 1st entry to the NE zone (**C**). The values are expressed as the means ± SEMs. ^#^
*p* < 0.001 vs. control group and * *p* < 0.05 vs. the MK-801-treated animals (N = at least 8–10).

**Figure 5 biomolecules-13-01064-f005:**
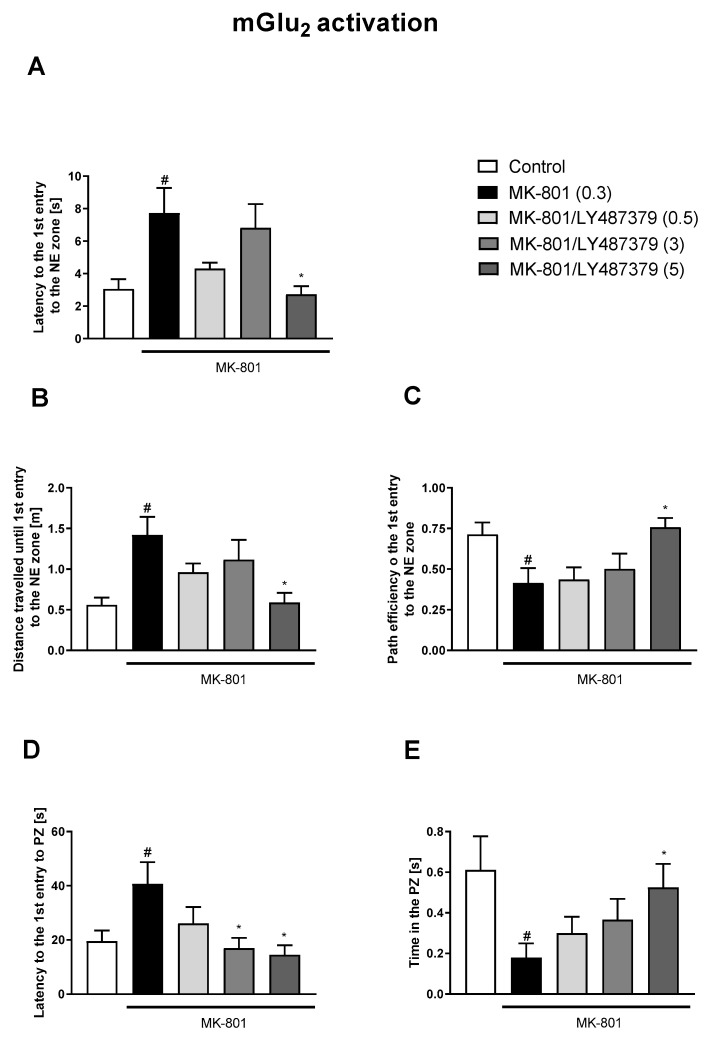
The effects of LY487379 on MK-801-induced deficits in the retention of spatial memory in the MWM in CD1 mice. Sub-figures represent latency to the 1st entry to the NE zone (**A**), distance travelled until 1st entry to the NE zone (**B**), path efficiency to the 1st entry to the NE zone (**C**), latency to the 1st entry to the PZ (**D**) and time spent in the PZ (**E**). The values are expressed as the means ± SEMs. ^#^
*p* < 0.001 vs. control group and * *p* < 0.05 vs. the MK-801-treated animals (N = at least 8–10). PZ—platform zone.

**Figure 6 biomolecules-13-01064-f006:**
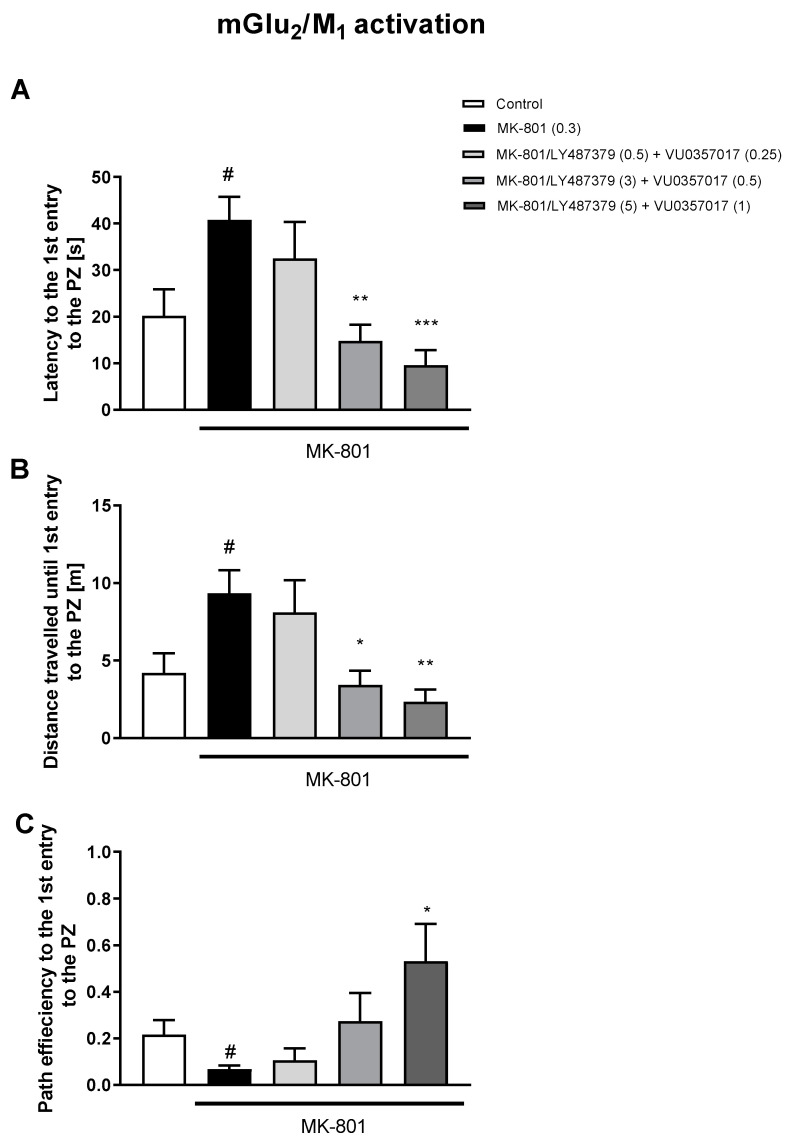
The effects of concomitant administration of LY487379 and VU0357017 on MK-801-induced deficits in the retention of spatial memory in the MWM in CD1 mice. Sub-figures represent latency to the 1st entry to the PZ (**A**), distance travelled until 1st entry to the PZ (**B**) and path efficiency to the 1st entry to the PZ (**C**). The values are expressed as the means ± SEMs. ^#^
*p* < 0.001 vs. control group and * *p* < 0.05, ** *p* < 0.01, and *** *p* < 0.001 vs. the MK-801-treated animals (N = at least 8–10). PZ—platform zone.

**Figure 7 biomolecules-13-01064-f007:**
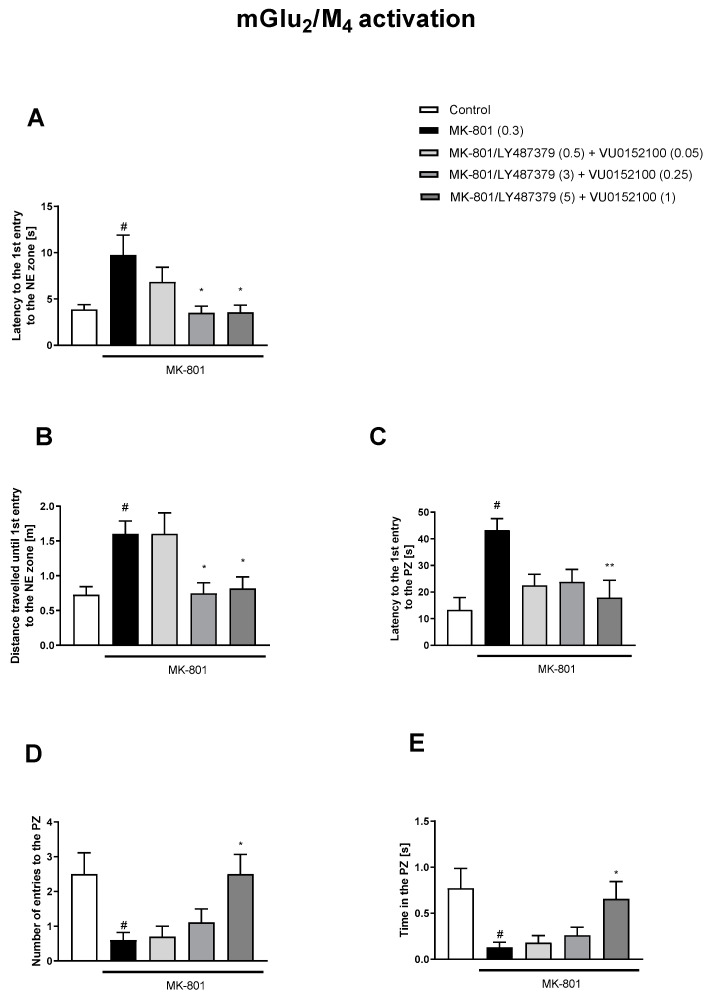
The effects of concomitant administration of LY487379 and VU015100 on MK-801-induced deficits in the retention of spatial memory in the MWM in CD1 mice. Sub-titles represent latency to the 1st entry to the NE zone (**A**), distance travelled until 1st entry to the NE zone (**B**), latency to the 1st entry to the PZ (**C**), number of entries to the PZ (**D**) and time spent in the PZ (**E**). The values are expressed as the means ± SEMs. ^#^
*p* < 0.001 vs. control group and * *p* < 0.05, ** *p* < 0.01, vs. the MK-801-treated animals (N = at least 8–10). PZ—platform zone.

**Figure 8 biomolecules-13-01064-f008:**
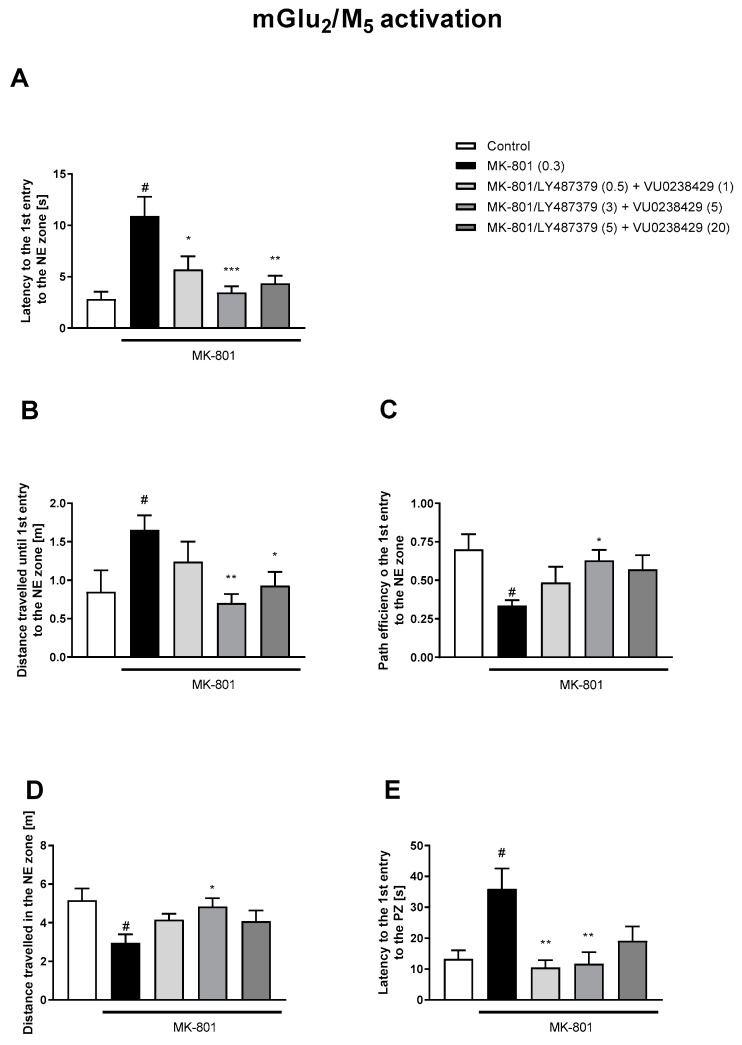
The effects of concomitant administration of LY487379 and VU0238429 on MK-801-induced deficits in the retention of spatial memory in the MWM in CD1 mice. Sub-titles represent latency to the 1st entry to the NE zone (**A**), distance travelled until 1st entry to the NE zone (**B**), path efficiency to the 1st entry to the NE zone (**C**), distance travelled in the NE zone (**D**) and latency to the 1st entry to the PZ (**E**). The values are expressed as the means ± SEMs. ^#^
*p* < 0.001 vs. control group and * *p* < 0.05, ** *p* < 0.01, vs. the MK-801-treated animals (N = at least 8–10). PZ—platform zone.

**Figure 9 biomolecules-13-01064-f009:**
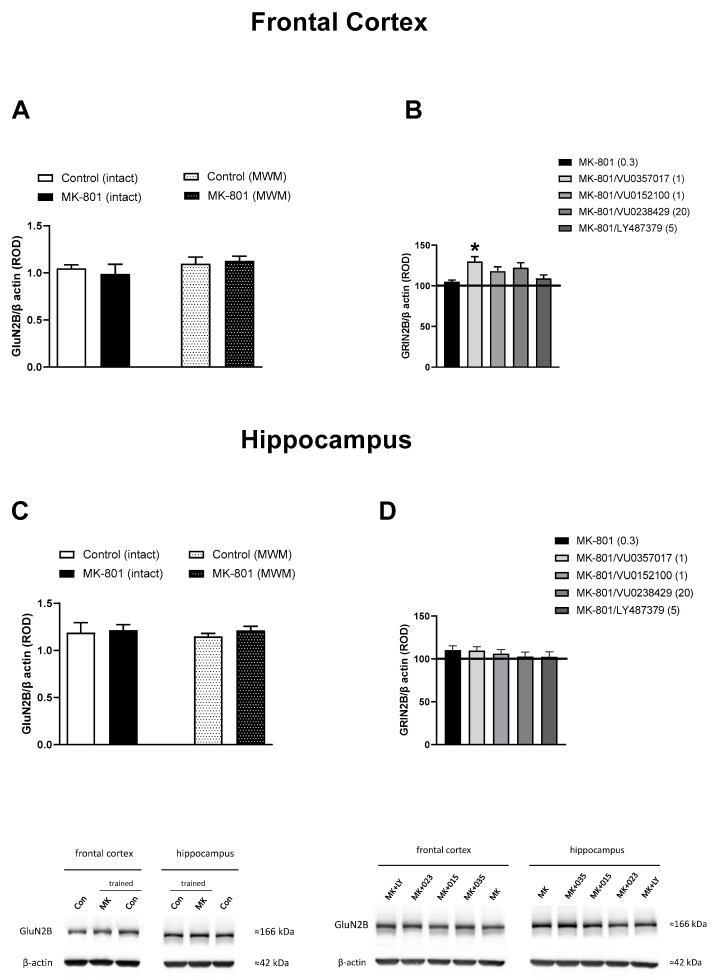
The effect of the Morris Water Maze (MWM) procedure on GluN2B expression in the frontal cortex (**A**) and hippocampus (**C**) of the mouse brain, and the effect of the investigated compounds on GluN2B expression in the frontal cortex (**B**) and hippocampus (**D**) of MWM-trained mice. ROD, relative optical density. The values are expressed as the means ± SEMs. * *p* < 0.05 vs. the MK-801-treated animals (N = at least 8–10).

**Figure 10 biomolecules-13-01064-f010:**
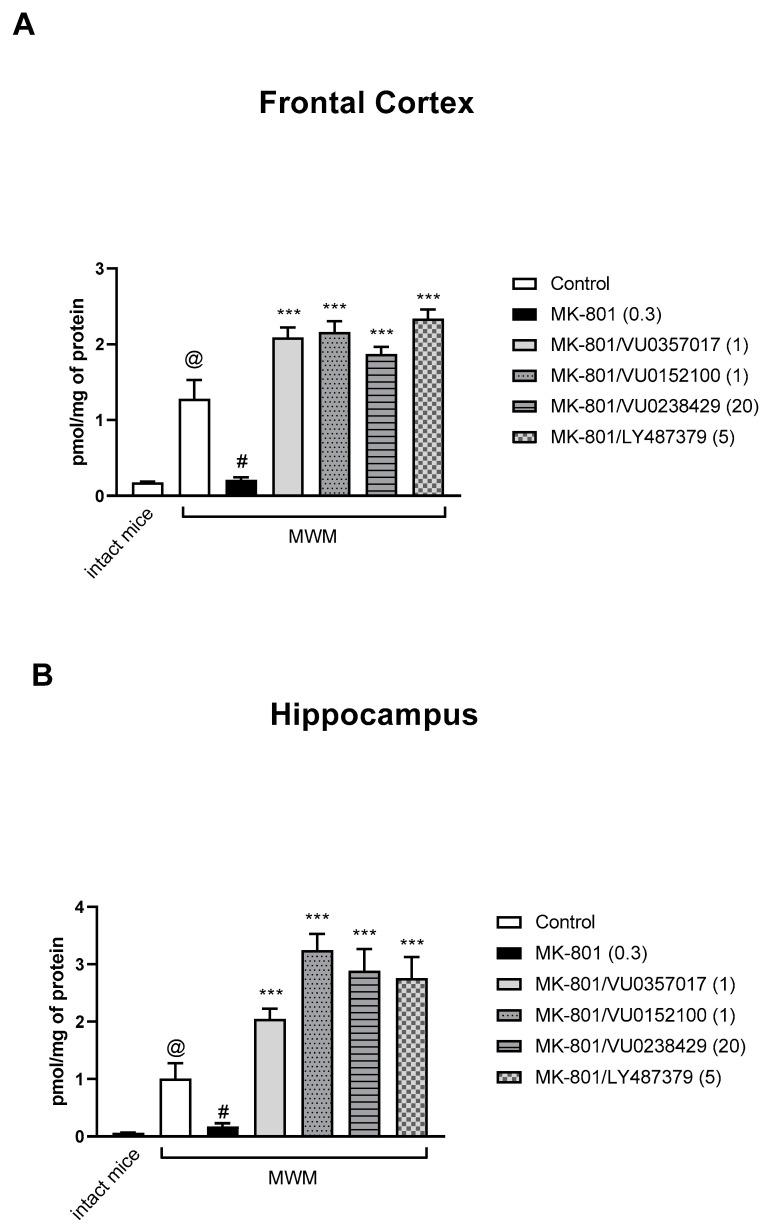
The effect of the Morris Water Maze (MWM) procedure on cGMP synthesis, and the effect of the investigated compounds in the frontal cortex (**A**) and hippocampus (**B**) of MWM-trained mice. The values are expressed as the means ± SEMs. @ *p* < 0.001 vs. intact mice, ^#^
*p* < 0.001 vs. MK-801-treated animals, and *** *p* < 0.0001 vs. the MK-801-treated animals (N = at least 8–10).

**Table 1 biomolecules-13-01064-t001:** Schematic representation of the doses of compounds administered in individual combinations.

**Inactive doses**		
		**LY487379**
**VU0357017**	0.25 mg/kg	0.5 mg/kg
**VU0152100**	0.05 mg/kg	0.5 mg/kg
**VU0238429**	1 mg/kg	0.5 mg/kg
**Intermediate doses**		
		**LY487379**
**VU0357017**	0.5 mg/kg	3 mg/kg
**VU0152100**	0.25 mg/kg	3 mg/kg
**VU0238429**	5 mg/kg	3 mg/kg
**Highest doses**		
		**LY487379**
**VU0357017**	1 mg/kg	5 mg/kg
**VU0152100**	1 mg/kg	5 mg/kg
**VU0238429**	20 mg/kg	5 mg/kg

## Data Availability

The data presented in this study are available on request from the corresponding author. The data are not publicly available due to limits.

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
