# Peer review of "Activation of Metabotropic Glutamate Receptor (mGlu2) and Muscarinic Receptors (M1, M4, and M5), Alone or in Combination, and Its Impact on the Acquisition and Retention of Learning in the Morris Water Maze, NMDA Expression and cGMP Synthesis"

_biomolecules, 2023, doi:10.3390/biom13071064_

Round 1
Reviewer 1 Report
The manuscript by Wieronska et al., assessed the ability of different muscarinic and metabotropic glutamate (mGlu2) receptor activators alone or in combination to counteract spatial navigation deficits induced by the NMDA receptor antagonist MK-801 in mice using the Morris water maze (MWM) test. Authors report that all the muscarinic and (mGlu2) receptor activators attenuated spatial memory deficits caused by MK-801. By contrast, the coadministration of the (mGlu2) compound LY487379 with each of the muscarinic activators failed to reverse MK-801 induced spatial learning and memory deficits. Further, MWM learning resulted in increased cGMP synthesis (cortex and hippocampi) which was prevented byMK-801. Finally, NMDA expression was not affected by treatment. The manuscript is a result of hard work, the findings appear interesting but also present some significant weaknesses.
Major concerns
1. Introduction
Please provide some more information regarding the involvement of muscarinic receptors in schizophrenia.
2. Experimental design
The experimental design is incomplete. Muscarinic and (mGlu2) activators control groups were not included in the study. Authors should provide, at least, appropriate information concerning the per se effects of these compounds on spatial learning.
3. Statistical analyses
The chapter of statistical analyses is not clear. How were acquisition data analyzed? A repeated measurement test (repetitive ANOVA) was used?
4. Swimming speeds
Did the authors find differences in swimming speeds among the various groups?
5. Lines 465 and 526. NOR it is not only “a short-term working memory test”. Based on the delay posed between the sample and the choice trial it can also considered as a long-term memory test. Further it is difficult to consider NOR as a “working memory test”. In this context, it has been reported that NOR can be only used to examine short and long-term memories or the memory of an episode (Ennaceur A., Object novelty recognition memory, page 2, 2018, In: Handbook of object novelty recognition).
6. Discussion
Authors correctly stated that “treatment did not affect NMDAR2B expression. Which is the biological significance of it?
7. Conclusions
Please summarize the results of the study. The first sentence of it is redundant.
Minor concerns
1. Line 64. Please specify that MK-801 is a NMDA receptor antagonist.
2. Material and methods
Please report the body weight and age of mice.
Author Response
Major concerns
- Introduction
Please provide some more information regarding the involvement of muscarinic receptors in schizophrenia.
The Introduction was reorganized also according to the remarks of the Reviewer #2. The information regarding muscarinic receptors in schizophrenia was incorporated. Lines 50-59
- Experimental design
The experimental design is incomplete. Muscarinic and (mGlu2) activators control groups were not included in the study. Authors should provide, at least, appropriate information concerning the per se effects of these compounds on spatial learning. Lines 214-215.
The appropriate information was included in the results section and in the Supplementary Materials Fig. 3.
- Statistical analyses
The chapter of statistical analyses is not clear. How were acquisition data analyzed? A repeated measurement test (repetitive ANOVA) was used?
Actually the statistical analysis were performed only on the last day of the training. Because of the large variations between the individuals in the group between particular trials and also variations between days, it was difficult to extract any statistical data for the whole week. The appropriate information was included in the description.
Statistical significance was determined by Student’s t-test (control vs. MK-801), one-way ANOVA followed by Tukey’s post hoc comparison (cGMP—“control” groups), or Dunnet’s post hoc comparison (MWM, cGMP—treatment). If the assumption of normal distribution was not met, the data were analyzed using U Mann–Whitney tests or Kruskal–Wallis tests. Regarding the acquisition phase, only data from the last day were analyzed using one-way ANOVA or Kruskal-Wallis test to assess if at the end of training procedure there were significant differences between MK-801 and treatment groups. The differences between control and MK-801 were analyzed using Student’s t-test or U Mann–Whitney test. The data were analyzed using TIBCO Statistica (v.13.3) or GraphPad Prism (v.9.4.1) and presented as mean ± SEM.
- Swimming speeds
Did the authors find differences in swimming speeds among the various groups?
No changes were observed in average swimming speed among experimental groups during retention trial. Please see Table 3 in Supplementary Material
The information concerning this was included in the text, lines 387-389
- Lines 465 and 526. NOR it is not only “a short-term working memory test”. Based on the delay posed between the sample and the choice trial it can also considered as a long-term memory test. Further it is difficult to consider NOR as a “working memory test”. In this context, it has been reported that NOR can be only used to examine short and long-term memories or the memory of an episode (Ennaceur A., Object novelty recognition memory, page 2, 2018, In: Handbook of object novelty recognition).
Thank for this important remark. Of course short- and long term memory can be measured in both methods, depending on the particular scheme. We included appropriate citation in the text and corrected the statement concerning NOR. Lines 392 and 464.
- Discussion
Authors correctly stated that “treatment did not affect NMDAR2B expression. Which is the biological significance of it?
Also in line with the remarks of reviewer #2 we improved this paragraph. Please, see lines 420-435 and the rest of the paragraph.
- Conclusions
Please summarize the results of the study. The first sentence of it is redundant.
Corrected.
Minor concerns
- Line 64. Please specify that MK-801 is a NMDA receptor antagonist.
Corrected. 2. Material and methods
Please report the body weight and age of mice.
Included in the line 75-78.
Thank you for the revision !

Reviewer 2 Report
Dear Authors,
Your manuscript entitled “Activation of metabotropic glutamate receptor (mGlu2) and muscarinic receptors (M1, M4, and M5), alone or in combination, and its impact on the acquisition and retention of learning in the Morris Water Maze, NMDA expression and cGMP synthesis” presents very interesting results of a well-designed experimental study focusing on cognitive function of mice.
First, I would like to appraise the methodology used and the meticulous analysis of the results, but I will also point out some concerns regarding the manuscript.
Overall, it is not very clear what was the main objective of the study. In the Introduction, the schizophrenia and antipsychotic drugs are emphasized, but the pharmacological substances used in this study are not in the drug development pipeline, or it is very unlikely to push them into this direction. On the other hand, the Discussion focuses on the findings and is structured very well, so I suggest to modify the Introduction and place the focus on the literature data regarding the principles and neurobiology of Morris water maze testing.
Specific comments: there are too many figures. Consider putting a part of them in supplementary files.
Please argument shortly why the NMDAR2B has been chosen. It is not discussed in detail, and the reference provided (Taylor et. al, 2014) presented data regarding GluN1 (GRIN1). Please use the new nomenclature: GRIN2B.
Minor errors only.
Author Response
Your manuscript entitled “Activation of metabotropic glutamate receptor (mGlu2) and muscarinic receptors (M1, M4, and M5), alone or in combination, and its impact on the acquisition and retention of learning in the Morris Water Maze, NMDA expression and cGMP synthesis” presents very interesting results of a well-designed experimental study focusing on cognitive function of mice.
First, I would like to appraise the methodology used and the meticulous analysis of the results, but I will also point out some concerns regarding the manuscript.
Overall, it is not very clear what was the main objective of the study. In the Introduction, the schizophrenia and antipsychotic drugs are emphasized, but the pharmacological substances used in this study are not in the drug development pipeline, or it is very unlikely to push them into this direction. On the other hand, the Discussion focuses on the findings and is structured very well, so I suggest to modify the Introduction and place the focus on the literature data regarding the principles and neurobiology of Morris water maze testing.
We reorganized Introduction section also taking into consideration remarks of the 1st reviewer who suggested to include more information regarding muscarinic receptors in schizophrenia.
Please see lines 62-70 concerning the principles of MWM test. The discussion also deeply focuses on this.
Specific comments: there are too many figures. Consider putting a part of them in supplementary files.
We put selected Tables and Figures into Supplementary Material, as suggested by the Reviewer.
Please argument shortly why the NMDAR2B has been chosen. It is not discussed in detail, and the reference provided (Taylor et. al, 2014) presented data regarding GluN1 (GRIN1). Please use the new nomenclature: GRIN2B.
Please see lines 420-447 in Discussion section which was improved accordingly.
Actually GRIN2B refers to the gene encoding the subunit. The new nomenclature regarding the protein is GluN2B, which was corrected across the manuscript.
Thank you for the revision.

Round 2
Reviewer 1 Report
Authors addressed all concerns raised. Manuscript now can be accpeted for publication.